# Associations between the Home Environment, Feeding Practices and Children’s Intakes of Fruit, Vegetables and Confectionary/Sugar-Sweetened Beverages

**DOI:** 10.3390/ijerph17134837

**Published:** 2020-07-05

**Authors:** Carolina Bassul, Clare A. Corish, John M. Kearney

**Affiliations:** 1School of Biological and Health Sciences, Technological University Dublin, City Campus, Kevin Street, Dublin 8, Ireland; 2School of Public Health, Physiotherapy and Sports Science, University College Dublin, Belfield, Dublin 4, Ireland; clare.corish@ucd.ie

**Keywords:** children’s diet, home environment, feeding practices

## Abstract

Within the home environment, parents influence their children’s dietary intakes through their parenting and dietary practices, and the foods they make available/accessible. The aim of this cross-sectional study was to examine the associations between home environmental characteristics and children’s dietary intakes. Three hundred and thirty-two children aged three–five years and their parents participated in the study. Home environmental characteristics, including parental control feeding practices, were explored using validated and standardized questionnaires such as the Child Feeding Questionnaire (CFQ), the Physical and Nutritional Home Environment Inventory (PNHEI) and the Healthy Home Survey (HHS). Parent and child food consumption was also measured. Pressure to eat from parents was associated with lower fruit intake in children (OR 0.67, 95% CI 0.47–0.96, *p* = 0.032). Greater variety of fruit available in the home increased the likelihood of fruit consumption in children (OR 1.35 95% CI 1.09–1.68, *p* = 0.005). Watching television for ≥1 h per day was associated with a decreased probability of children eating vegetables daily (OR 0.38, 95% CI 0.20–0.72, *p* = 0.003) and doubled their likelihood of consuming confectionary/sugar-sweetened beverages more than once weekly (OR 2.15, 95% CI 1.06–4.38, *p* = 0.034). Children whose parents had lower vegetable consumption were 59% less likely to eat vegetables daily. This study demonstrates that modifiable home environmental characteristics are significantly associated with children’s dietary intakes.

## 1. Introduction

The development of children’s dietary habits involves many interacting factors within different domains. During the life course, the home environment is acknowledged as an important setting in shaping children’s diets [1,2]. The home environment can be characterized by sociodemographic, behavioral and physical domains. A large cross-European (Belgium, Bulgaria, Germany, Greece, Poland and Spain) study of pre-schoolers (3.3–5.5 years old) and their mothers that examined differences in dietary quality among pre-schoolers according to gender, weight status and mothers’ socioeconomic status (SES) provided evidence that SES, predominantly determined by mothers’ education, is positively correlated with children’s higher diet quality scores [3]. The study authors hypothesized that higher education level in mothers positively influences their beliefs, knowledge and motivations around their child’s diet [3].

Parental behavioral characteristics such as feeding practices also play an important role in the development of children’s food preferences, eating habits and body weight [4,5]. Feeding practices are defined as the behavioral strategies that parents use to influence their children’s food intake [6]. These include monitoring food intake, restrictive or controlled feeding, pressurizing the child to eat and instrumental and emotional feeding [4,5,7,8,9]. For example, controlled or restrictive feeding practices have been shown to have a strong positive association with high child BMI in several cross-sectional studies [10,11,12,13,14,15]. In contrast, putting pressure on a child to eat has been associated with lower BMI [12,16,17,18,19,20] and child appetite characteristics such as food fussiness and slow eating [21,22]. Suboptimal feeding practices may disrupt children’s self-regulation of hunger and satiety, which may contribute to the development of overweight and obesity in young children [23].

In the case of dietary habits, feeding experience in early childhood can shape children’s subsequent dietary intake [24]. Studies have demonstrated that breastfeeding duration is positively associated with better quality of diet, with higher intakes of fruits and vegetables among pre-school children observed [25]. In contrast, early introduction of complementary foods (e.g., <17 weeks) has been associated with poor dietary quality, overweight and obesity in children [26]. In later childhood, parents may also shape their children’s eating behavior not only by the food choices they make for their family but by their own eating habits [27,28,29]. Several studies show the positive impact of parental role modelling on children’s eating habits, mainly on their consumption of fruits and vegetables [2,30,31,32]. Family mealtimes are also an excellent opportunity to establish healthy eating habits in early childhood. During meals, parents can make healthy options available, give children autonomy and the opportunity for food choice and exploration, and also encourage a positive family dynamic and environment [33,34]. For example, negative interactions between parents and children, such as restriction of foods or negative statements about a child’s food intake, are associated with higher weight status in children [35,36], while eating family meals while watching television is associated with consumption of unhealthy foods [37,38,39,40]. Conversely, interpersonal dynamics such as warmth and group enjoyment during the family meal are associated with lower risk of child overweight or obesity, higher fruit and vegetable intakes and increased frequency of family meals [34,41]. 

The contribution of the physical domain of the home environment on children’s dietary habits and weight status has also been examined [42]. For example, a recent cross-sectional study demonstrated a positive association between the availability and accessibility of vegetables in the home and higher frequency of vegetable consumption among three–five-year-old children [43]. A recent systematic review of older children (aged 6–12 years old) also reported consistent evidence of children’s intake of fruits and vegetables in the home being positively related to availability and accessibility of these foods. The authors highlighted the importance of both factors (accessibility and availability) as children may not consume fruits and vegetables that are inaccessible, even if available, especially if unhealthy foods are also available or accessible [44].

To date in Ireland, there is little research using an ecological approach to understand the influence of the home environment and parental feeding practices on pre-school children’s dietary behaviors. In this context, the present study increases our knowledge of the influence of important home environmental factors (parents’ attitudes towards children’s diet, parents’ own dietary intakes, family mealtimes, parents’ rules and polices around children’s screen time and home foods availability/accessibility) on pre-school children’s dietary intakes. Such knowledge will inform public health interventions and provide health professionals with accurate information that will allow them to effectively advise how healthier home environment for this age group can be created. Developing effective interventions to promote healthier patterns throughout the life course requires an understanding of the dietary behaviors of young children and the mediating influences of parenting practices and home environmental factors that influence early health behaviors [45]. Therefore, this study aimed to examine the associations between home environmental characteristics, including parental control feeding practices and children’s intakes of fruit, vegetables and confectionary/sugar-sweetened beverages (SSBs).

## 2. Materials and Methods 

### 2.1. Participants and Study Procedures 

The study sample comprises 332 children aged three–five years old and their parents/guardians (herein referred to as ‘parents’). The participants were recruited through a randomized stratified sampling of pre-schools in different geo-socioeconomic areas of Dublin, Ireland. To meet the study inclusion criteria, the pre-school had to provide a structured pre-school service for a minimum of 3 h and be registered with the Irish Child and Family Agency (TUSLA). The pre-school managers/owners (*n* = 85) were first contacted via posted leaflet, then email and telephone call, inviting them to participate in the study following a detailed explanation of what the study was about. Interested pre-schools were then compiled into an “agreed” list *(n* = 25), pre-schools not interested in participating were classified as “not interested” (*n* = 24) and managers/owners who were unable to be reached by telephone were classified as “failed to contact” (*n* = 36). From the 25 participating pre-schools, all children aged three–five years old, free from any chronic medical condition that affected their growth and development, and their parents were eligible for inclusion in the study. Parents with insufficient English language proficiency to complete the questionnaire were excluded. A total of 670 parents received the study questionnaire pack provided by pre-school staff. This pack contained an information letter, consent forms (study consent form and a separate anthropometric measurement consent form) and the “Pre-schoolers health study” questionnaire. The parents were given one week to return the completed questionnaire to the pre-school in a sealed envelope. A reminder was sent to those who had not returned the questionnaire pack after one week *(n =* 440) and a further week was given for them to return the information. After two weeks, the researcher (CB) collected the sealed questionnaire packs. Data collection started in August 2016 and was completed in June 2017 with data being collected in two–three pre-schools per month over this 10-month period. The study was approved by the Ethics Committee of the Technological University Dublin (TUD) (Ref 15–109).

### 2.2. Study Instrument: “Pre-Schoolers Health Study” Questionnaire

#### 2.2.1. General Information 

The participants’ characteristics were taken from the general information section of the “Pre-schoolers health study” questionnaire. This section assessed the sociodemographic and general health information of the parent completing the questionnaire and the child referred to in the questionnaire. These characteristics included gender (parent and child), parent’s age, parent’s self-reported weight and height, education level, nationality and marital status. 

Child’s breastfeeding was assessed by the questions: “Was your child breastfed?” Parents answers were coded as “ever breastfed or never breastfed.” The end of exclusive breastfeeding was defined as the age of introduction of any infant formula and then coded as <6 months or ≥6 months in accordance with the WHO recommendation for exclusive breastfeeding [46]. The timing of introduction of complementary foods was assessed by the question “how old was your child when she/he was first introduced to solid food?” Parents responded in weeks or months. The introduction of complementary foods was dichotomized as <17 weeks or ≥17 weeks, consistent with previous studies of infants and pre-school children, and Irish recommendations [26,47]. The childcare arrangements were also recorded. 

#### 2.2.2. Children’s and Parents’ Food Consumption 

Children’s dietary intake was determined by markers of a healthy (fruit and vegetables) or unhealthy (confectionary/SSBs) diet [48] and were assessed through the question “over the past month how often does your child eat or drink…” Possible responses ranged from 1–6 ((1) 3 or more times a day, (2) 1 or 2 times a day, (3) 4 to 6 times per week, (4) 1 to 2 times per week, (5) less than once per week and (6) never). Fruit and vegetable intakes were assessed separately and dichotomized as (0) <1 serving a day or (1) ≥1 serving a day. Such dichotomization has been used in previous National and European studies [49,50]. Intake of confectionary/SSBs was dichotomized as (0) <1 time per week or (1) ≥1 time per week based on the Irish Food and Nutrition Guidelines for pre-school children [51]. The parents’ food consumption questionnaire was similar to that used to assess the dietary intake of their children with the foods assessed similar to those assessed in their children: fruit, vegetables, snacks such as biscuits, cake, chocolates and crisps as well as SSBs. Fruit and vegetable consumption were assessed separately and dichotomized as ≥3 servings a day or <3 servings daily. Parents’ snack and SSB consumption were dichotomized as ≥3 times or <3 times per week. 

#### 2.2.3. Parental Control Feeding Practice 

The parental control feeding practice was assessed using 10 items from the validated Child Feeding Questionnaire (CFQ) [7]. Parental restrictive feeding practice measured the extent to which parents restrict their child’s access to foods (four items, e.g., “I have to be sure that my child does not eat too many sweets” and “I intentionally keep some foods out of my child’s reach”). Monitoring feeding practice was used to evaluate how much parents oversee their child’s eating (three items, e.g., “How much do you keep track of sweets such as candy, ice-cream, cakes and pastries that your child eats”). Parental pressure to eat assessed the parents’ tendency to pressurize their children to eat more food (three items, e.g., “If my child says ‘I am not hungry’ I try to get him/her to eat anyway”). The variable score for each controlling feeding practice assessed (restriction, monitoring and pressure to eat) was obtained by calculating the mean score for the items on each variable [7]. 

#### 2.2.4. Family Mealtime 

Family mealtimes were measured using a previously used standardized questionnaire The Physical and Nutritional Home Environment Inventory (PNHEI) [52]. Parents reported the daily time spent cooking; how many days per week meals were prepared at home; how many meals the child eats with one or both parents; type of food consumed during meals (takeaways, microwavable and quick-cook frozen foods); how often the child helps to prepare food and the mealtime environment. For comparative purposes, mealtime variables were dichotomized as follows: time spent cooking: <30 min or ≥30 min; meals prepared at home: <5 times per week or ≥5 times per week; main meal with one or both parents: <5 times per week or ≥5 times per week [53]. Parents responded to each statement using a four-point Likert scale (frequently, sometimes, occasionally and rarely/never) that was subsequently dichotomized as frequently or rarely/never.

#### 2.2.5. Children’s Television Viewing 

To establish children’s television viewing, parents responded to the statement “On average how many hours per day does your child watch any type of television including DVD and videos?” Response options were (1) none, (2) less than 30 min a day, (3) 30 min to 1 h a day, (4) 1 to 1 ½ h a day, (5) 1 ½ to 2 h a day, (6) more than 2 h a day. The responses were then dichotomized as <1 h daily or ≥1 h daily, consistent with recent World Health Organization (WHO) recommendations on screen time for children under five years old [54]. Children’s television viewing while eating was assessed through a question “how often does your child eat meal (or snack) in front of the television (turned on)? Parents responded on four-point Likert scale (frequently, sometimes, occasionally and rarely/never), which was subsequently dichotomized as frequently or rarely/never. 

#### 2.2.6. Home Food Availability and Accessibility

The validated questionnaire Healthy Home Survey (HHS) was used to measure home food availability and accessibility [54]. For this study, food availability was based on the variety of foods present in the home; for example, the number of different types of foods within each category; fruit (fresh, dried, frozen and canned fruits); vegetables (fresh, frozen and canned vegetables); confectionary (e.g., biscuits, cakes/muffins and pastries); drinks (e.g., fizzy drinks, sugar-sweetened juices and squashes/cordials); savory snacks (e.g., chips, tortillas, peanuts and cheese puffs). To evaluate the child’s accessibility to food in the home, parents were asked: “would it be possible for your child to get any foods by him/herself without your help? (by this we mean whether it would be physically possible?). If yes, which foods? (e.g., fruit, vegetables, confectionary, drinks and savory snacks).” Parents were also asked whether their child was allowed to take these foods without their consent.

### 2.3. Statistical Analysis

IBM SPSS for macOS Mojave, version 26.0 (IBM, New York, NY, United States) was used to analyze data. The statistical significance level was set at *p* < 0.05 for all analyses. Normally distributed data and non-normally distributed data were summarized using the mean and standard deviation (SD) and the median and interquartile range (IQR), respectively. Categorical variables were dichotomized into two or three categories and presented both as the number in each category *(n*) and as a percentage (%) of the total. Initial bivariate associations between children’s dietary intake (categorical) and home environmental characteristics were undertaken. Normally distributed continuous variables were analyzed using an Independent Samples t-test, non-normally distributed variables using the Mann-Whitney U test. Categorical variables were assessed using cross-tabulations with the Chi-squared test used to assess statistical significance.” Yates’ Continuity Correction was used for 2 × 2 contingency tables to improve the Chi-square approximation [55]. Multivariate analysis of variance (MANOVA) was performed to investigate differences between parents’ education level and parental control feeding practices after controlling for the sociodemographic characteristics associated with the control feeding practices in the Independent Samples t-test. Multicollinearity was also assessed before multivariate analyses were conducted. To avoid multicollinearity, a multiple linear regression analysis for the set of independent variables was conducted. In the multiple linear regression analysis, the multicollinearity was checked by run collinearity and measured by Tolerance <0.10 and a Variance Inflation Factor (VIF) >10. No multicollinearity was identified within the independent variables.

Binary logistic regression was then performed on home environmental characteristic variables significantly associated with children’s dietary intake in the bivariate analysis. The dependent variables were coded as: fruit 0 = <1 serving a day or 1 = ≥1 serving a day; vegetables 0 = <1 serving a day or 1 = ≥1 serving a day and confectionary/SSBs 0 = <1 time per week or 1 = ≥1 time per week. The independent variables (participants’ characteristics, family mealtimes, children’s television viewing and home food availability) were dummy categorized. The Forced Entry Method was used, whereby the significant variables in the unadjusted analysis were tested in one block to assess their association with the dependent variables. Once the association analysis had been performed, the usefulness of each model was assessed [56]. The Hosmer-Lemeshow Goodness of Fit Test was used to ensure that *p* was <0.05, to indicate support for the model. The Cox and Snell R Square and the Nagelkerke R Square values provided information about the usefulness of the model; these indicate the amount of variation in the outcome variable which can be explained by the model [56]. The significance value produced by the Wald test for each independent variable in the model was then checked. Variables with a significance of *p* < 0.05 contributed significantly to the associative ability of the model and, therefore, were significantly associated with the dependent variables. For comparison purposes, the unadjusted and adjusted exponentiation of the B coefficient-Exp (B) value or odds radios (OR) and 95% confidence interval (CI) were also recorded for each independent variable.

## 3. Results

### 3.1. Characteristics of Participants 

Three hundred and thirty-two child–parent dyads were included in the analysis. An overview of participants’ characteristics is provided in Table 1. All children were aged 5.5 or less years (mean 4.4, SD 0.8), with boys and girls almost equally represented, 50.2%, *n =* 165 boys. Most children were ever breastfed (66.3%, *n =* 220); however, the duration of exclusive breastfeeding was less than that recommended by the WHO (70.9%, *n* 205, <6 months) [46]. Introduction of complementary foods was established in compliance with the Irish recommendations by almost three-quarters of parents (74.8%, *n =* 237, ≥17 weeks) [47]. All children attended pre-school, most for less than 5 h daily (63.6%, *n =* 210). 

The majority of parents who participated in this study were mothers (88.3%, *n =* 293), with an age range of 30–39 years (63.6%, *n =* 211), married or living together with their partners (81.3%, *n =* 270) and of Irish nationality (66.9%, *n =* 222). Over half of the parents (56.6%, *n =* 188) were educated to higher degree level, either undergraduate or postgraduate. At the time of the study, 38.1% (*n =* 115) of parents/guardians were overweight or obese.

### 3.2. Participants’ Characteristics Associated with Parental Control Feeding Practices 

Parents tended to exert more restrictive (mean 4.2, SD 0.82), and monitoring (mean 4.2, SD 0.94) feeding practices and put less pressure in their child to eat foods, typically at mealtimes (mean 2.9, SD 1.02). The bivariate analysis of children’s characteristics and parental control feeding practice demonstrated that children’s age, gender and BMI were not associated with any parental control feeding practice. Irish parents tended to exert more restriction and monitoring compared to non-Irish parents (Table 2). Pressure to eat score was significantly higher among parents from a lower level of education after parents’ nationality was controlled for (Appendix A). 

### 3.3. Participants’ Behavioral Characteristics and Home Food Availability and Accessibility 

Table 3 outlines participants’ behavioral characteristics for family mealtimes and children’s television viewing. Eating with their children was common practice for the majority of parents (66.9%, *n =* 222). Although most parents reported cooking more than five days weekly and spending over 30 min cooking, more than half reported frequent consumption of takeaway foods, which children also ate (52.1% *n =* 171), and one-third used microwavable and quick-cook frozen foods (32.5%, *n =* 108). Despite the majority of parents reporting that they set rules on children’s television viewing (79.2%, *n* = 283), eating meals and snacks while watching television was a common practice among 52.1%, *n* = 173 and 78.3%, *n =* 260 of children, respectively. Over half (56%, *n =* 186) of children watched more than one hour of television daily, higher than that recommended by WHO [54]. 

Regards to home food availability almost all participants reported having fruit at home with a median of 5.0, IQR 4.0–6.0 types available (Figure 1). A similar number of participants reported having vegetables at home; however, the median score was higher than for fruit (median 7.0, IQR 5.0–8.0). Again, a majority reported having sweet snacks at home with a median 2.0, IQR 2.0–3.0 types available. Food accessibility was indicated by the presence of foods on display, whether children were able to reach the foods without help and whether they were allowed take the foods by themselves without their parents’ consent (Figure 1). Fruit was the most accessible food for both being able to reach and allowed to take, 74%, *n =* 242 and 34.3%, *n* = 108 of children, respectively.

### 3.4. Home Environmental Characteristics Associated with Children’s Dietary Intake

#### 3.4.1. Children’s Fruit Intake

Table 4 shows the unadjusted and adjusted models resulting from all home environmental characteristic variables associated with children’s fruit intake. In the unadjusted analysis, children’s fruit intake was associated with parental characteristics such as education level, BMI and fruit, vegetable and SSB intake (Appendix A). From the participants’ behavioral characteristics, three variables related to children’s television viewing were negatively associated with children’s fruit intake (*p* < 0.05). Fruit and vegetable availability in the home increased the likelihood of higher fruit consumption in children, by 14% and 11%, respectively. 

In the adjusted model, two of the 14 independent variables included made a statistically significant contribution to the model. Fruit availability in the home was independently associated with children’s fruit intake. Having a greater variety of fruits available increased by 35% the likelihood of children’s fruit consumption (OR 1.35, 95% CI 1.09–1.68, *p* = 0.005). Children whose parents put pressure on them to eat were 33% less likely to consume fruits daily compared to children whose parents exerted less pressure on them to eat (OR 0.67, 95% CI 0.47–0.96, *p* = 0.032).

#### 3.4.2. Children’s Vegetable Intake

Table 5 outlines the home environmental characteristics that were associated with children’s vegetable intake in the unadjusted and adjusted models. Twenty one of 42 variables showed an association with children’s vegetable intake in the unadjusted analysis (Appendix A). Similarly, to the relationships observed with children’s fruit intake, parental higher education showed a positive association with children’s vegetable intake while parents’ higher BMI was negatively associated with children’s consumption of vegetables (*p* = 0.010). 

Consideration of participants’ behavioral characteristics indicated that all four variables related to children’s television viewing were negatively associated with children’s vegetable consumption. In the adjusted analysis, the strongest home environmental characteristic associated with children’s vegetable intake was found to be eating snacks while watching television. Children whose parents allowed snacks to be eaten in front of the television were 71% less likely to consume vegetables daily compared with those whose parents did not allow snacks to be eaten in front of the television (OR 0.29, 95% CI 0.12–0.71, *p* = 0.006). A negative association between children’s daily television viewing and children’s vegetable consumption was also observed (OR 0.38, 95% CI 0.20–0.72, *p* = 0.003). Parents’ vegetable intake showed a strong positive correlation with children’s vegetable intake; children whose parents consumed fewer than three servings of vegetables daily were 59% less likely to consume vegetables daily compared to children whose parents consumed more than three servings of vegetables per day.

#### 3.4.3. Children’s confectionary/SSB intakes

Results from the unadjusted and adjusted models of the home environmental characteristic variables associated with children’s confectionary/SSB intakes are provided in Table 6. In the unadjusted analysis, participants’ characteristics associated with children’s confectionary/SSB intakes were parents’ nationality, parental education, parents’ BMI and timing of introduction of complementary foods. Parents’ vegetable consumption showed a negative association with children’s confectionary/SSB intakes (Appendix A). Watching over one hour of television daily and eating while watching television were positively associated with children’s likelihood of consuming confectionary/SSBs weekly (*p* < 0.05). Greater variety of vegetables available at home was associated with a 17% lower chance of children consuming confectionary/SSBs weekly. In contrast, availability and accessibility of foods from the top shelf of the Food Pyramid were positively correlated with children’s confectionary/SSB intakes.

Only four of the 23 independent variables included made a statistically significant contribution to the model. The results showed that introduction of complementary foods established before 17 weeks was associated with four times the probability of children consuming confectionary/SSBs more than once per week when three–five years old (OR 4.02, 95% CI 1.42–11.35, *p* = 0.009). Children who watched more than one hour of television daily doubled their chance of consuming confectionary/SSBs weekly compared to children who watched less than one hour of television daily (OR 2.15, 95% CI 1.06–4.38, *p* = 0.034). Home food availability was also found to be independently associated with children’s confectionary/SSB intakes. The higher the number of different types of vegetables in the home, the lower the likelihood of children consuming confectionary/SSBs weekly (by 19%, OR 0.81, 95% CI 0.69–0.95, *p* = 0.014). Conversely, a greater variety of SSBs available was associated with a 63% increased chance of children consuming confectionary/SSBs ≥ 1 time per week (OR 1.63, 95% CI 1.06–2.49, *p* = 0.023).

## 4. Discussion

This study examined the home environmental characteristics associated with children’s fruit, vegetable and confectionary/SSB intakes. The data demonstrated that children consumed almost twice as much fresh fruit (excluding fruit juice, canned or dried fruits) than vegetables. Such a pattern of fruit and vegetable consumption among infants and pre-schoolers has previously been observed in national and international studies [57,58]. Consumption of biscuits, chocolates, muffins, sweets and SSBs on a weekly basis was common practice for over 70% of pre-schoolers in this study. This figure may be influenced by the offer of “treat” foods to children, which has been shown to be a habitual practice in Ireland [59,60]. A recent Irish study of adult treat-giving behaviors to children reported that unhealthy foods such as sweets and chocolates were the most common “treats” offered (57.8%). For 57.2% of adults, treats were offered to children routinely, between one–four times per week [59]. It is important to consider that intake of individual food items cannot fully represent the habitual diet. However, the role of fresh fruit and vegetables in a healthy diet is clear and, moreover, has been shown to be a good marker for a heathy diet [49]. Furthermore, excessive consumption of foods and drinks high in fat and sugar increase the risk of childhood obesity and other diet-related conditions such as dental caries and weight-related comorbidities [61].

Parental characteristics such as education level and BMI were consistently associated with children’s consumption of fruits, vegetables and confectionery/SSBs. Higher level of parental education was associated with higher intake of fruits and vegetables and lower intakes of confectionary/SSBs in children. Our findings are supported by recently published data from a European study involving 10 countries, which associated parents’ education with fruit and vegetable consumption in 11-year-old children [50]. The children of more highly educated parents had higher fruit intakes in five out of 10 countries, while the association between higher parental education and higher daily vegetable consumption in children was reported in seven out of 10 countries. Associations between confectionary/SSB intakes were not investigated in this international study [50]. However, in another study, parents with lower education had children with a higher intake of snack foods, SSBs and lower quality diet in childhood and adolescence [62]. Education may reflect parents’ capacity to access, interpret and put into practice health information. For instance, parents with higher education have been shown to make healthy options more available and accessible to children [63]. Parents’ BMI is an indicator of environmental and sociocultural factors and is associated with children’s dietary intake and weight status in previous studies [64,65]. In our study, children whose parents were overweight/obese were less likely to consume fruit and vegetables, and more likely to have confectionary/SSBs one or more times per week, compared to children whose parents were normal weight. The results of the study by Wardle et al. concur with our results; this study demonstrated that children aged 4–5 years old with overweight/obese parents were less likely to consume vegetables compared to children with normal weight parents [66]. 

Our study demonstrated that earlier introduction of complementary foods (<17 weeks) was positively associated with higher confectionary/SSB intakes in the unadjusted and adjusted models. An association between the timing of introduction of complementary foods was established and children’s dietary patterns and overweight/obesity has been examined in several studies with contradictory findings [67,68]. Results from a cross-sectional study which looked at dietary risk among children aged one–five years, breastfeeding duration and age of introduction to solids, demonstrated that poor diet quality at five years was related to shorter duration of breastfeeding, but failed to find an association with age of introduction of solids. Earlier introduction of complementary foods has been positively associated with poor dietary quality in children, with higher consumption of foods high in fat and SSBs in two previous studies [69,70]. In addition, a nationally representative longitudinal study ‘Growing up in Ireland’ reported that 3-year-old children who had been introduced to complementary foods after 17 weeks were less likely to be overweight or obese [71].

In pre-school aged children, dietary habits are heavily influenced by observational learning; consequently, parents’ diets are likely to impact on their children’s dietary habits [72,73]. Our study found that not all foods from parents’ and children’s diets were positively associated. The most consistent findings we observed were for parent-child vegetable intakes. International studies which investigated dietary intakes in parent-child dyads have shown mixed results, with weak associations and variation by nutrient, food and parent-child dyad (such as mother-daughter or parent-offspring) [72]. A meta-analysis by Wang et al. established that similarities between parents’ and children’s dietary intakes focused on total energy and dietary fat and concluded that the associations were weak [74]. Overall, mixed results in this field might be explained by the fact that parents may influence their children’s eating habits not only by their own eating habits but also by providing a food environment that encourages healthy or unhealthy eating habits [75].

The parental control feeding practice, ‘pressure to eat’ was positively associated with children’s lower fruit intake. Pressure to eat is normally used to encourage children’s food consumption [76]. It has been hypothesized that pressure to eat is associated with food preferences, fussy eating, less healthy diet and lower BMI [8]. Indeed, a review of the literature concluded that children whose mothers put more pressure on them to eat had lower consumption of fruits and vegetables and higher overall fat intake; these associations were observed independently of family socioeconomic status [77]. A systematic review of the literature supports the association between parental pressure to eat and children’s lower BMI in 11 cross-sectional studies, one longitudinal study and one randomized control trial (RCT). The authors attempted to explain their findings by stating that pressure to eat may have been applied in response to children’s lower BMI and that it may not be the cause of the lower BMI [18]. In the longitudinal study, the authors assessed the directionality in relation to fussy eating and pressure to eat. The study findings showed a positive bi-directional relationship; pre-schoolers’ fussy eating predicted parental pressure to eat at four years of age. Parental pressure to eat at four years of age predicted children’s eating fussiness at six years [78]. 

The present study showed a strong association between watching more than one hour of television daily and lower vegetable consumption and higher confectionary/SSB intakes in children. International studies have identified associations between children’s television viewing, diet quality and overweight/obesity [37,79,80]. Television may influence what children eat through advertisement of energy-dense foods and promotion of mindless eating during viewing, which, in turn, may lead to overeating and overweight/obesity [39]. Indeed, a recent systematic review, which included 19 observational studies, reported that all studies found a significant positive correlation between television viewing and/or total screen time and unhealthy dietary behaviors such as lower fruit and vegetable intakes and greater consumption of foods higher in fat and sugar [81]. Such findings are in line with the present study in which children who eat snacks while watching television were over 70% less likely to eat vegetables daily. Based on these findings, and taking into consideration that in pre-school aged children, the family environment remains a major contributor to the development of children’s eating behavior, it is advised that parents offer healthier snacks, such as fruit bowls or vegetable platters in an attractive way to children while watching television and to restrict television viewing during meal and snack times [81]. 

Home food availability and accessibility are among the determinants that are most consistently associated with dietary intake in children and adolescents [82,83,84]. In the present study, the adjusted model showed that the availability of fruits, vegetables and SSBs in the home were associated with higher children’s fruit and confectionary/SSB intakes. Such findings are consistent with a cross-sectional study which examined the home food environment and quality of diet in parent–child dyads in which greater availability of healthful foods was associated with a higher diet quality score [85]. Another international study with pre-schoolers reported that having foods high in sugar in the home was associated with unhealthy dietary patterns, even when fruit and vegetables were also available [82]. A literature review and meta-analysis supported the view that the availability of food items was consistently associated with healthy or unhealthy food consumption in children [83]. The authors hypothesized that it is possible that food availability might increase consumption by children since they tend to eat whatever is available to them [83]. 

Home food accessibility is determined by whether foods are available in a form and location that facilitates their consumption; for instance, if a food item is located in a visible place such as the kitchen counter [86]. In the present study, food accessibility was measured through food visibility—the food being easy for children to reach without adult help, and whether parents allowed children to serve themselves without asking permission. The number of parents that allowed children to take foods without permission was very low, thus, making it difficult to conduct a statistical analysis with adequate power. This is probably due to the age of the study population, since children under five years are usually not allowed to take food without their parents’ or caregivers’ permission. Thus, in the present study, the only food accessibility variable that showed a relationship with children’s dietary intake was related to food visibility. A previous study with pre-school children, which examined fruit and vegetable storage in a ready-to-eat form and children’s fruit and vegetable consumption, demonstrated that parents were more likely to feed their pre-school children with foods that are more accessible [2]. Overall, healthy and/or unhealthy food availability and accessibility alone or in combination have been associated with children’s intake of such foods. This is, therefore, an important issue that must be taken into consideration in the development of interventions that could provide home environments favorable to healthy eating [1,84,87]. 

Before drawing conclusions, the strengths and limitations of this study must be considered. This is one of the first studies in Ireland to provide insights into the relationship between home environmental characteristics and pre-school children’s dietary intakes. As home environment and children’s diet have been found to be related to socioeconomic status, an important strength of this study is the representation (over 45%) of parents of lower education, particularly as such parents have been under-represented in previous studies [43]. A further strength of our study is the use of validated instruments, which allows for direct comparison with previous studies in the same age group. 

The limitations of this study must also be considered. The study design is cross-sectional, hence, it can only demonstrate associations between the variables investigated but cannot demonstrate cause and effect. Furthermore, although the response rate in this study was over 50%, the results are not nationally representative. Moreover, the assessment of diet is known to be challenging and parents’ ability to recall their children’s diet may be a limiting factor. Administration of the self-completion questionnaires used in this study may have introduced positive response bias. The measurement of the home environment is complex with many dynamic influences; therefore, it is also possible that important family environment variables have been involuntarily excluded or poorly characterized.

## 5. Conclusions 

The present study corroborates the findings of previous studies by clearly demonstrating that different components of the home environment were associated with children’s intakes of fruits, vegetables and confectionary/SSBs. For example, the parental control feeding practice—pressure to eat—was positively associated with lower fruit intake in children. Considering that feeding practice is a modifiable factor associated with children’s dietary intake, further understanding the differences in use of these practices by parents of differing sociodemographic characteristics could be beneficial. Some similarities between parents’ and children’s diets were observed; interventions focused on improving parental diet may contribute, in part, to children’s healthier dietary intakes. The associations between children’s television viewing and markers of healthy and unhealthy diets such as fruit, vegetable and confectionery/SSB intakes were particularly strong. Findings from this study highlight the importance of limiting screen time, especially during meal and snack times. Home food availability appears to have an important impact on children’s diets, since this is associated with higher fruit and lower confectionary/SSB intakes. The holistic approach taken in our study provides a valuable contribution to the published literature on pre-school children’s dietary intakes. Examining the effects of different factors within the home environment is important to the provision of accurate information that can be used to improve children’s dietary intakes and, consequently, overall health. It is anticipated that findings from the present study could inform the development and implementation of intervention strategies as well as help health professionals to work with parents and pre-school care providers to ensure healthier diets for this age group.

## Figures and Tables

**Figure 1 ijerph-17-04837-f001:**
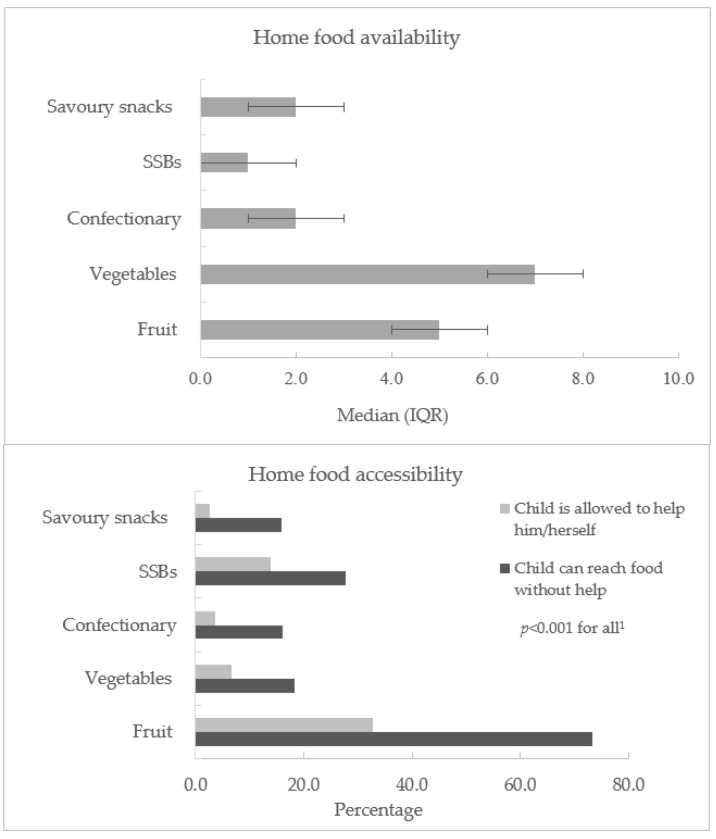
Home food availability and accessibility. IQR: Interquartile range; SSB: sugar-sweetened beverage; ^1^ Association between food accessibility variables assessed using the chi-squared test with Yates’ Continuity Correction for 2 × 2 contingency tables.

**Table 1 ijerph-17-04837-t001:** Characteristics of participants.

Characteristics		*n*	%	Mean	SD
**Children**					
Age (*n* = 332)				4.4	0.81
Gender (*n* = 329)	Female	164	49.8		
	Male	165	50.2		
Child z-BMI-age (*n =* 204)				0.74	0.92
Breastfeeding (*n* = 331)	Ever breastfed	220	66.3		
	Breastfed exclusively				
	<6 months	205	70.9		
	≥6months	84	29.1		
	Never breastfed	111	33.7		
Introduction of complementary foods (*n =* 317)	<17 weeks	80	25.2		
	≥17weeks	237	74.8		
Attending pre-school (*n =* 330)	Full-time	120	36.4		
	Part-time	210	63.6		
Fruit intake (*n =* 332)	≥1 serving a day	235	70.8		
	<1 serving a day	97	29.2		
Vegetable intake *(n =* 332)	≥1 serving a day	129	38.9		
	<1 serving a day	203	61.1		
Confectionary/ SSB intake (*n* = 332)	≥1 time per week	241	72.6		
	≥time per week	91	27.4		
**Parents**					
Age (*n* = 332)	20–29 years	40	12		
	30–39 years	211	63.6		
	40 or more years	81	24.4		
Relationship with child (*n =* 332)	Mother	293	88.3		
	Father	33	9.9		
	Other	6	1.8		
Nationality (*n* = 332)	Irish	222	66.9		
Marital status (*n* = 332)	Married/living together	270	81.3		
Education level^1^ (*n =* 332)	Higher	188	56.6		
	Lower	144	43.4		
Household income (*n* = 263)	<40.000 €/p.a.	86	32.7		
	≥40.000 €/p.a.	177	67.3		
Parents’ BMI (*n =* 302)	Normal weight	187	61.9		
	Overweight/obese	115	38.1		
Fruit intake (*n =* 332)	≥3 servings a day	77	23.2		
	<3 servings a day	255	76.8		
Vegetable intake *(n =* 332)	≥3 servings a day	109	32.8		
	<3 servings a day	223	67.2		
Confectionary/savory snack intake *(n =* 332)	≥3 times per week	159	47.9		
	<3 times per week	173	52.1		
SSB intake *(n* = 332)	≥1 time per week	77	23.2		
	<1 time per week	255	76.8		

BMI: Body mass index, SD: Standard deviation, SSB: Sugar-sweetened beverage. p.a.: per annum. ^1^ Higher education includes undergraduate and post-graduate education, lower education is secondary school or less, further education covers education and training that occurs after second level schooling, but which is not part of the third level system.

**Table 2 ijerph-17-04837-t002:** Participants’ sociodemographic characteristics associated with parental control feeding practice.

		Parental Control Feeding Practice								
		Restriction Range 1–5)			Monitoring (Range 1–5)			Pressure (Range 1–5)		
Characteristics		Mean	SD	*p **	Mean	SD	*p **	Mean	SD	*p **
**Children**										
Age ^2^		4.22	0.82	0.287	4.27	0.94	0.704	2.93	1.02	0.725
Gender ^3^	Female	4.30	0.76	0.176	4.36	0.88	0.117	2.96	0.96	0.695
	Male	4.18	0.86		4.19	0.99		2.92	1.07	
Child z-BMI-age ^2^		4.22	0.82	0.143	4.27	0.94	0.209	2.93	1.02	0.270
**Parents**										
Age ^4^	20–29 years	4.03	1.01	0.190	4.19	1.03	0.658	3.21	1.08	0.169
	30–39 years	4.28	0.77		4.26	0.97		2.88	0.99	
	40 or more years	4.21	0.81		4.35	0.82		2.97	1.03	
Nationality ^3^	Not Irish	4.04	0.90	0.002	4.00	1.18	<0.001	3.09	1.08	0.067
	Irish	4.33	0.76		4.40	0.78		2.87	0.98	
Marital status ^3^	Married/living together	4.23	0.81	0.876	4.30	0.90	0.179	2.92	0.99	0.377
	Not married /living together	4.25	0.85		4.12	1.12		3.04	1.11	
Education level ^1,3^	Higher	4.28	0.78	0.302	4.29	0.94	0.741	2.80	0.99	0.004
	Lower	4.18	0.85		4.25	0.95		3.12	1.02	

SD: Standard deviation; ^1^ Higher education includes undergraduate and post-graduate education, lower education is secondary school or less, further education covers education and training that occurs after second level schooling but which is not part of the third level system; * *p* < 0.05 was significant; ^2^ Pearson correlation; ^3^ Association between normally distributed continuous data assessed using an Independent Samples *t*-test.

**Table 3 ijerph-17-04837-t003:** Participants’ behavioral characteristics.

Characteristics		*n*	%
Time spent cooking (*n =* 332)	<30 min	117	35.2
	≥30 min	215	64.8
Prepare meals at home (*n =* 332)	<5 times per week	61	18.4
	≥5 times per week	271	81.6
Main meal with one or both parents (*n =* 329)	<5 times per week	107	32.2
	≥5 times per week	222	66.9
Microwavable or quick-cook frozen foods consumed (*n* =329)	Frequently	108	32.5
	Rarely/never	221	66.6
Takeaway food which the child also eats (*n =* 328)	Frequently	171	52.1
	Rarely/never	157	47.9
Child helps with food preparation (*n =* 329)	Frequently	181	55.0
	Rarely/never	148	45.0
Relaxing meal environment (*n* = 332)	Very relaxing	61	18.4
	Comfortable	186	56.0
	Not relaxing	85	25.6
Parents set rules about television viewing (*n =* 330)	Yes	263	79.2

Children’s daily television viewing (*n =* 332)	<1 h daily	146	44.0
	≥1 h daily	186	56.0
Parents allow meal to be eaten in front of television (*n =* 330)	Frequently	173	52.1
	Rarely/never	157	47.3
Parents allow snacks to be eaten in front of television (*n* = 329)	Frequently	260	78.3
	Rarely/never	69	20.8

**Table 4 ijerph-17-04837-t004:** Home environmental characteristics associated with children’s fruit intake.

		Fruit			
		Unadjusted		Adjusted	
Characteristics		OR (95% CI)	*p**	OR (95% CI)	*p **
Education level	Higher ^(Ref.)^				
	Lower	0.38 (0.23–0.63)	<0.001	0.73 (0.34–1.53)	0.409
Parents’ BMI	Normal weight ^(Ref.)^				
	Overweight/obese	0.51 (0.31–0.85)	0.010	0.76 (0.39–1.49)	0.440
Parents’ fruit intake	≥3 servings a day ^(Ref.)^				
	<3 servings a day	0.41 (0.21–0.79)	0.007	0.38 (0.13–1.08)	0.072
Parents’ vegetable intake	≥3 servings a day ^(Ref.)^				
	< 3 servings a day	0.46 (0.26–0.08)	0.005	1.14 (0.50–2.56)	0.750
Parents’ SSB intake	≥1 time per week	0.41 (0.24–0.70)	0.001	0.83 (0.37–1.85)	0.832
	<1 time per week ^(Ref.)^				
Children’s daily television viewing	<1 h daily ^(Ref.)^				
	≥ 1 h daily	0.49 (0.29–0.80)	0.005	0.99 (0.47–2.08)	0.996
Parents set rules about television viewing	No	0.57 (0.32–1.00)	0.005	0.57 (0.26–1.25)	0.168

Parents allow snacks to be eaten in front of television	Frequently	0.44 (0.22–0.88)	0.018	0.76 (0.31–1.87)	0.559
	Rarely/never ^(Ref.)^				
Parental control feeding practice	Pressure	0.69 (0.54–0.88)	0.002	0.67 (0.47–0.96)	0.032
	Monitoring	1.31 (1.03–1.67)	0.012	1.28 (0.88–1.86)	0.192
Home food availability	Fruit types	1.48 (1.26–1.73)	<0.001	1.35 (1.09–1.68)	0.005
	Vegetable types	1.15 (1.04–1.27)	0.007	1.01 (0.87- 1.17)	0.866
Home food accessibility (child can reach food without help)					
Sweet snacks	Yes	0.41 (0.22–0.76)	0.004	0.71 (0.28–1.82)	0.482
SSBs	Yes	0.48 (0.29–0.81)	0.005	1.42 (0.67–2.98)	0.354

Adjusted model summary: *x*^2^ (13, n 298) = 58.42, *p* < 0.001; correctly predicted 76.8% of cases and explained between 17% (Cox and Snell R square) and 25% (Nagelkerke R square) of the variance. CI: Confidential Interval; OR: Odds Ratio; Ref.: Reference group; * *p* < 0.05 was significant.

**Table 5 ijerph-17-04837-t005:** Home environmental characteristics associated with children’s vegetable intake.

		Vegetables			
		Unadjusted		Adjusted	
Characteristics		OR (95% CI)	*p **	OR (95% CI)	*p **
Education level	Higher ^(Ref.)^				
	Lower	0.40 (0.25–0.64)	<0.001	0.67 (0.34–1.30)	0.240
Parents’ BMI	Normal weight ^(Ref.)^				
	Overweight/obese	0.57 (0.35–0.93)	0.010	0.61 (0.32–1.17)	0.139
Introduction of complementary foods	<17 weeks	0.58 (0.33–0.99)	0.046	1.19 (0.56–0.56)	0.639
	≥17 weeks ^(Ref.)^				
Parents’ vegetable intake	≥3 servings a day ^(Ref.)^				
	<3 servings a day	0.27 (0.16–0.44)	0.001	0.41 (0.21–0.82)	0.012
Parents prepare meals at home	<5 times per week	0.54 (0.29–1.0)	0.050	0.87 (0.37–2.02)	0.748
	≥5 times per week ^(Ref.)^				
Eat main meal with one or both parents	<5 times per week	0.57 (0.34–0.93)	0.025	0.65 (0.33–1.30)	0.229
	≥5 times per week ^(Ref.)^				
Microwavable or quick-cook frozen foods consumed	Frequently	0.54 (0.33–0.89)	0.016	0.80 (0.39–1.65)	0.557
	Rarely/never ^(Ref.)^				
Takeaway food which the child also eats	Frequently	0.63 (0.40–0.99)	0.048	1.34 (0.69–2.60)	0.373
	Rarely/never ^(Ref.)^				
Children’s daily television viewing	<1 h daily ^(Ref.)^				
	≥1 h daily	0.34 (0.22–0.54)	<0.001	0.38 (0.20–0.72)	0.003
Parents set rules about television viewing	No	0.45 (0.25–0.83)	0.011	0.77 (0.34–1.74)	0.543

Parents allow meal to be eaten in front of television	Frequently	0.41 (0.26–0.64)	<0.001	1.17 (0.58–2.35)	0.647
	Rarely/never ^(Ref.)^				
Parents allow snacks to be eaten in front of television	Frequently	0.19 (0.10–0.34)	<0.001	0.29 (0.12–0.71)	0.006
	Rarely/never ^(Ref.)^				
Parental control feeding practice	Pressure	0.60 (0.47–0.75)	<0.001	0.74 (0.54–1.01)	0.059
	Monitoring	1.37 (1.06–1.77)	0.002	1.20 (0.84–1.71)	0.296
Home food availability	Fruit types	1.35 (1.18–1.55)	<0.001	1.18 (0.96–1.45)	0.112
	Vegetables types	1.20 (1.09–1.33)	<0.001	1.12 (0.97–1.30)	0.116
	Sweet snacks types	0.85 (0.72–0.99)	0.032	0.88 (0.69–1.14)	0.359
	Sugary drinks	0.73 (0.59–0.92)	0.007	0.92 (0.67–1.26)	0.620
	Savory snacks	0.76 (0.62–0.92)	0.004	0.81 (0.61–1.08)	0.162
Home food accessibility (child can reach food without help)					
Fruit	Yes	1.88(1.10–3.22)	0.020	1.36 (0.65–2.85)	0.411
Sweet snacks	Yes	0.44(0.22–0.88)	0.018	0.86 (0.34–2.20)	0.763

Adjusted model summary: x^2^ (15, n330) =92.827, *p* <0.001. The model correctly predicted 73.7% of cases and explained between 29% (Cox and Snell R square) and 39% (Nagelkerke R square) of the variance. CI: Confidential Interval; OR: Odds Ratio; Ref.: Reference group; * *p* < 0.05 was significant.

**Table 6 ijerph-17-04837-t006:** Home environmental characteristics associated with children’s confectionary/SSB intakes.

		Confectionary/SSBs			
		Unadjusted		Adjusted	
Characteristics		OR (95% CI)	*p **	OR (95% CI)	*p **
Education level	Higher ^(Ref.)^				
	Lower	2.57 (1.52–4.35)	<0.001	0.70 (0.32–1.55)	0.384
Nationality	Irish ^(Ref.)^				
	Not Irish	0.59 (0.36–0.97)	0.040	0.79 (0.38–1.67)	0.542
Parents’ BMI	Normal weight ^(Ref.)^				
	Overweight/obese	2.03 (1.17–3.51)	0.010	1.74 (0.83–3.65)	0.140
Introduction of complementary foods	<17 weeks	4.67 (2.14–10.17)	<0.001	4.02 (1.42 -11.35)	0.009
	≥17 weeks ^(Ref.)^				
Parents’ vegetable intake	≥3 servings a day ^(Ref.)^				
	<3 servings a day	1.95 (1.18–3.22)	0.008	1.10 (0.52–2.34)	0.803
Parent’s confectionary/ savory snacks intake	≥3 times per week	2.19 (1.32–3.62)	0.002	1.84 (0.90–3.73)	0.093
	<3 times per week ^(Ref.)^				
Parent’s SSB intake	≥1 time per week	4.91 (2.16–11.15)	<0.001	2.03 (0.72–5.67)	0.179
	<1 time per week ^(Ref.)^				
Microwavable or quick-cook frozen foods consumed	Frequently	1.88 (1.08–3.27)	0.024	0.80 (0.37–1.73)	0.566
	Rarely/never ^(Ref.)^				
Takeaway food which the child also eats	Frequently	2.71 (1.64–4.50)	<0.001	1.98 (0.98–4.03)	0.059
	Rarely/never ^(Ref.)^				
Children’s daily television viewing	<1 h daily ^(Ref.)^				
	≥1 h daily	2.22 (1.35–3.60)	0.001	2.15 (1.06–4.38)	0.034
Parents allow snacks to be eaten in front of television	Frequently	3.53 (2.02–6.17)	<0.001	1.32 (0.61–2.89)	0.481
	Rarely/never ^(Ref.)^				
Home food availability	Vegetable types	0.83 (0.75–0.92)	0.001	0.81 (0.69–0.95)	0.014
	Sweet snacks types	1.46 (1.20–1.76)	<0.001	1.21 (0.93–1.59)	0.162
	Sweet drink types	2.37 (1.72–3.26)	<0.001	1.63 (1.06–2.49)	0.023
	Savory snack types	1.36 (1.09–1.70)	0.001	1.06 (0.80–1.42)	0.681
Home food accessibility (child can reach food without help)					
Vegetables	No	2.26 (1.26–4.07)	0.005	0.51 (0.22–1.19)	0.117
Sweet snacks	Yes	2.61 (1.13–6.05)	0.038	0.88 (0.29–2.66)	0.820
SSBs	Yes	2.39 (1.29–4.43)	0.009	0.56 (0.22–1.47)	0.241

Adjusted model summary: *x*^2^ (18, n 271) = 85.662, *p* < 0.001, the model correctly predicted 81.5% of cases and explained between 27.1% (Cox and Snell R square) and 39.4% (Nagelkerke R square) of the variance. CI: Confidential Interval; OR: Odds Ratio; Ref.: Reference group; * *p* < 0.05 was significant.

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
