# Peer review of "Associations between the Home Environment, Feeding Practices and Children’s Intakes of Fruit, Vegetables and Confectionary/Sugar-Sweetened Beverages"

_ijerph, 2020, doi:10.3390/ijerph17134837_

Round 1
Reviewer 1 Report
This paper examines the associations between different components of the home environment and children’s dietary intake of fruits, vegetables and confectionary/sugar-sweetened beverages (SSB). Results indicate parental education level, BMI and dietary habits are strongly associated with children’s healthier/unhealthier dietary intakes. Furthermore, earlier weaning, parental control feeding practices, such as “pressure to eat”, TV screen time and food availability influence children’s eating behavior toward fruits, vegetables and confectionary/SSBs.
The article is based on a well prepared and executed research. The goals and rationale of the study
are clearly stated in the paper. The analysis and calculations are well executed, utilizing the relevant statistical methodology. The presentation of the results is good, and discussions of results is quite robust.
A minor comment: Check table 3 – the columns margins, because the last digit of p-value is lost in some rows!
Overall, the article is an interesting read and it is my recommendation it should be published.
Reviewer 2 Report
This well-conducted and well-written study aimed to investigate factors associated with children’s intake of fruit, vegetables, and confectionary/sugar-sweetened beverages. The authors included a variety of possible determinants, which make this study very interesting and important.
However, I have several minor comments:
1. Abstract – please, add the short names of used questionnaires
2. Line 38: please, add the reference for this particular study
3. Introduction – please add some information about early-feeding factors (breastfeeding and complementary feeding) related to subsequent children’s dietary intake, especially that you also included those variables in your study)
4. Methods – there is no information about breastfeeding. I also include transferring the part with references from results (lines 222-223) to the method section.
5. Methods and Results (lines220-223) – there is no information on which type of breastfeeding did you breastfeeding. Any, ever, or exclusive? You wrote that 66% of children were breastfed, but 41% were breastfed for less than 6 months, despite that WHO recommends the 6 months of exclusive breastfeeding. Does this mean that 24.7% were breastfed exclusively for longer than 6 months?! Moreover, in table 1a please add the option “never breastfed”.The numerousness in this variable is not clear: 220 of children were ever breastfed, whereas in table 1 you wrote n = 331, but later on you present results for only 220 kids – please unify these results (eg. adding the category 'never breastfed'). Moreover, please check the WHO recommendations about complementary feeding which is recommended around 26 week, not >=17… Also, in my opinion, if you refer to WHO recommendations you should cite them despite the Irish one. Moreover, I suggest changing the term “weaning” with “introducing to complementary foods”, because weaning referring more to stopping breastfeeding than introducing solids (especially that WHO recommends the breastfeeding continuation up to 2 years and beyond alongside with solids).
6. Please describe other nationalities. Did all the parents of different nationalities were well English-speaking?
7. I think that table S1 and S2 is quite important and should be transferred to the main manuscript. I suggest removing the Figure 1.
8. Figure 2b – please add some p-values.
9. I also suggest writing “n = xxx” instead of “n xxx”.
Reviewer 3 Report
This study aims to provide a greater understanding of the influences, particularly related to parents and the home environment, on dietary habits of pre-school age children in Ireland. The authors provide an extensive analysis of these various factors. For the most part, the description of the analysis and factors considered is thorough and clear. There are just a few points that need to be addressed.
Methods
All of the surveys are well explained except the home food availability and accessibility; this section is very vague. Given that you are providing detail to the results, further explanation of this survey would be helpful.
Results
Line 362: “top self”…? Please fix or explain.
Discussion
Line 386: Add “on” after SSB.
Line 418: Add “of” after timing.
Reviewer 4 Report
The introduction section is well written and I really enjoyed reading it with clear literature review. The paper is very interesting and really deal with a relevant topic. However, in the introduction part the jump to the objective was abrupt. I mean the author mentioned the gap that want to fill (the key question is clear) but it is not motivated why this specific question against other in children eating behavior? and which additional contribution is making to literature compared to other studies that analyze home environment.
This comment also come from the conclusion as the authors just comment that this study corroborates previous studies. So, the need to highlight a differentiated take-home key message is needed to put in evidence the added value of this research that I am sure it has.
In this context, while all the information presented in the introduction and later discussed is interesting, it would helpful to draw a theoretical scheme relating children eating behavior with the different variables and aspects identified from the literature review.
This recommendation turns to be important after have reading the whole paper. There is a lot of interesting outcomes and information and from the beginning putting the reader in a more holistic view of the research is helpful.
Why the data collection lasted 10 months? In the sample and data collection seem it last one month (A reminder was sent to those who had not returned the questionnaire pack after one week (n 440) and a further week was given for them to return the information. After two weeks…
Sample size and data collection. I am not sure regarding the journal rules. Was the questionnaire approved by an ethical committee mainly in relation to personal data collection, etc..
After reading the different part of the Pre-schoolers health study, I expected to see some “economic “information collected, since the authors demonstrated the significance of this typoe of variables (food expenditure, percentage on household income, if both parents are working, something related to social class…). Again is relevant to see the different part re-schoolers health questionnaire in a schematic way
Analysis. This sentence is confusing (Associations between normally distributed continuous data were explored using an Independent Samples t-test). If both variable are continuous and normally distributed, I expect to see the Pearson coefficient of association. Please clarify
More details should be given regarding the logit model. While this model is well known and any reader can look for details in literature, some additional information (empirical equation of the study) could be given (highlighting why is important to report adjusted and non-adjusted results and outcome). The independent variable (0 , 1) is not commented as such. And should mentioned in this part and also highlight that all the independent variable will by dummy categorized (as shown in the results tables).
Include the technical sheet of survey (confidence level and interval…)
88.3% of the parent sample were the mother and only 9.9 were the father (any reason for this outcome since the questionnaire should be answered at home and both can contribute). Thus, the affirmation “The bivariate analysis demonstrated that there were no significant gender differences in any parental control feeding practice” could be related more to facto of the small father sample size rather than non-significant association?
